# Dissociation, Cognitive Reflection and Health Literacy Have a Modest Effect on Belief in Conspiracy Theories about COVID-19

**DOI:** 10.3390/ijerph18105065

**Published:** 2021-05-11

**Authors:** Vojtech Pisl, Jan Volavka, Edita Chvojkova, Katerina Cechova, Gabriela Kavalirova, Jan Vevera

**Affiliations:** 1Department of Psychiatry, Faculty of Medicine and University Hospital in Pilsen, Charles University, 30100 Pilsen, Czech Republic; pisl@mail.muni.cz (V.P.); janvolavka@gmail.com (J.V.); 2Department of Psychiatry, School of Medicine, New York University, New York, NY 10016, USA; 3Department of Psychological Methods, University of Amsterdam, 1012 Amsterdam, The Netherlands; edita.chvojkova@student.uva.nl; 4Memory Clinic, Department of Neurology, Second Faculty of Medicine, Charles University and Motol University Hospital, 15006 Prague, Czech Republic; cechova.katerina@gmail.com; 5International Clinical Research Center, St. Anne’s University Hospital Brno, 65691 Brno, Czech Republic; 6Center of Physical Education and Sport, Faculty of Education, University of West Bohemia, 30100 Pilsen, Czech Republic; kavaliro@ktv.zcu.cz; 7Department of Psychiatry, First Faculty of Medicine, Charles University in Prague and General University Hospital in Prague, 12808 Prague, Czech Republic; 8Institute for Postgraduate Medical Education, 10005 Prague, Czech Republic

**Keywords:** conspiracy theories, COVID-19, health literacy, dissociation, cognitive reflection, bullshit receptivity, eHEALS

## Abstract

Understanding the predictors of belief in COVID-related conspiracy theories and willingness to get vaccinated against COVID-19 may aid the resolution of current and future pandemics. We investigate how psychological and cognitive characteristics influence general conspiracy mentality and COVID-related conspiracy theories. A cross-sectional study was conducted based on data from an online survey of a sample of Czech university students (*n* = 866) collected in January 2021, using multivariate linear regression and mediation analysis. Sixteen percent of respondents believed that COVID-19 is a hoax, and 17% believed that COVID-19 was intentionally created by humans. Seven percent of the variance of the hoax theory and 10% of the variance of the creation theory was explained by (in descending order of relevance) low cognitive reflection, low digital health literacy, high experience with dissociation and, to some extent, high bullshit receptivity. Belief in COVID-related conspiracy theories depended less on psychological and cognitive variables compared to conspiracy mentality (16% of the variance explained). The effect of digital health literacy on belief in COVID-related theories was moderated by cognitive reflection. Belief in conspiracy theories related to COVID-19 was influenced by experience with dissociation, cognitive reflection, digital health literacy and bullshit receptivity.

## 1. Introduction

The COVID-19 pandemic is accompanied and complicated by an associated infodemics-overabundance of information, making it difficult for the public to find relevant and correct information [1]. Among other misleading information, a wide range of conspiracy theories about COVID-19 are influencing public opinions, with some of them being credible for about 8% to 38% of the population in Western [2] and Eastern [3] Europe or Northern America [4], affecting adherence to anti-pandemic measures [3,5]. Susceptibility to conspiracy theories may be increased by dissociation: disintegration of experiences reducing awareness of intolerable information [6]. Belief in COVID-related conspiracy theories is predicted by cognitive reflection, the ability to reflect upon whether the result of an intuitive cognitive process is correct [7,8]. Beliefs in generic conspiracy theories are predicted by bullshit receptivity, the ability to judge whether a statement was “constructed absent direct concern for the truth” [9] (p. 549), even when controlled for cognitive reflection [10].

Misinformation about COVID-19 is decreased by the ability to access, understand, appraise and apply information about coronavirus [11], and digital health literacy reduces belief in COVID-related conspiracy [12]. For convenience, self-assessment questionnaires are often used, despite not being well-suited for interindividual comparisons [13]. In case respondents overestimate their digital health literacy, it may be especially relevant in those susceptible to conspiracy theories, as overestimating one’s own explanations contributes to both the risk of inflating one’s skills in a subjective health literacy questionnaire [13] and conspiratorial thinking [14]. Indeed, a meta-analysis revealed a mismatch between subjective and objective information literacy [15] and some studies led to counterintuitive results, such as a positive correlation between digital health literacy and conspiracy theories [16,17] or a negative correlation between digital health literacy and vaccination compliance [18]. Because digital literacy should, by definition [19], result in finding and applying health-related information in a way that helps to solve health problems, rather than increasing vulnerability to contagious diseases, we find these findings suspicious.

The presented study will examine dissociative tendencies, cognitive reflection, bullshit receptivity and digital health literacy as potential predictors of conspiracy mentality and belief in conspiracy theories about COVID-19 among university students. We expect that high dissociative tendencies, low cognitive reflection and high bullshit receptivity will predict high conspiracy mentality (hypothesis 1), that high dissociative tendencies, low cognitive reflection, high bullshit receptivity and low digital health literacy will predict belief in conspiracy theories (hypothesis 2), that conspiracy mentality will mediate the link between dissociative tendencies, cognitive reflection and high bullshit receptivity on one side and belief in COVID-related conspiracies on the other (hypothesis 3) and that the effects of dissociation and digital health literacy on COVID-related conspiracy beliefs will interact (i.e., in the effect of dissociation will be even stronger in individuals with low digital health literacy; hypothesis 4). Further, we want to address possible bias in eHEALS results due to overestimation of skills during self-evaluation. Due to the difficulties inherent to measuring digital health literacy objectively [13], we will examine this possibility through the effects of cognitive reflection and bullshit receptivity on the link between the eHEALS score and belief in conspiracy theories: We expect cognitive reflection (hypothesis 5) and bullshit receptivity (hypothesis 6) to moderate the link between digital health literacy measured by the self-reporting eHEALS questionnaire and belief in COVID-related conspiracy theories, indicating bias in the eHEALS results. If eHEALS is an unbiased measure of digital health literacy, neither of these characteristics should influence the relation between digital health literacy and belief in conspiracy theories: digital health literacy should predict lower belief in conspiracy theories related to health in people with low as well as high cognitive reflection and bullshit receptivity. However, eHEALS score may be inflated in people with low cognitive reflection (i.e., the respondents exaggerate their skills due to lack of awareness of their mistakes) or high bullshit receptivity (i.e., the respondents exaggerate their skills due to lack of ability to assess the actual quality of the information they trust). In such a case, cognitive reflection (and/or bullshit receptivity) will moderate the effect of eHEALS score on belief in conspiracy theories. Namely, in participants with low cognitive reflection (and/or high bullshit receptivity), the negative effect of digital health literacy on belief in health-related conspiracy theories will be weakened by the inflated eHEALS score.

## 2. Materials and Methods

### 2.1. Materials

Experience with dissociation (DES) was measured by the validated Czech version of the Dissociative experience scale [20], consisting of 28 items offering participants eleven options (0%, 10%, 20% … 100%) to mark how often they experience typical dissociative situations (e.g., missing a part of conversation or feeling as if one’s body was not one’s own). The resulting score is the mean of the scores for individual items. The assessment of internal reliability revealed good internal consistency (Cronbach alpha = 0.93).

Conspiracy mentality (CM) was measured by the Conspiracy Mentality Questionnaire (CMQ), a short instrument with high correlation with longer and more established Conspiracy Mentality Scale [21]. CMQ consists of five items, offering participants eleven options (0%, 10%, 20% … 100%) to mark their agreement with general conspiracy claims (such as “I think that government agencies closely monitor all citizens” or “I think that there are secret organizations that greatly influence political decisions”). The Czech translation of the original scale was confirmed by back translation. The resulting score is the mean of the scores for individual items. The assessment of internal reliability revealed good internal consistency (Cronbach alpha = 0.82).

The belief in two COVID-related conspiracy theories, namely, that COVID-19 is a hoax, and that COVID-19 was created intentionally by humans, was measured by two scales, each consisting of three items, adopted from [5]. Participants marked the probability that the three presented statements are true, on the same scale as above (0%, 10%, 20% … 100%). The hoax scale (HOAX) had good reliability in the previous research (Cronbach alpha = 0.85), consisting of three items: “The virus is intentionally presented as dangerous in order to mislead the public”, “Experts intentionally mislead us for their own benefit, even though the virus is not worse than a flu”, and “We should believe experts when they say that the virus is dangerous” (reverse-coded). The human-made scale (CREATED) shown adequate reliability given the low number of items (Cronbach alpha = 0.67) in the previous study [5], consisting of claims “Corona was intentionally brought into the world to reduce the population”, “Dark forces want to use the virus to rule the world”, and “I think it’s nonsense that the virus was created in a laboratory” (reverse-coded). The resulting scores for both dimensions are the means of the scores for individual items (except for those reverse-scored). The Czech translation of the original scale was confirmed by back translation and the assessment of internal reliability has shown consistencies similar to those in the original study (Cronbach alpha = 0.88 for HOAX and Cronbach alpha = 0.67 for CREATED).

Cognitive reflection (CRT) was measured by the cognitive reflection test [22], a short test that was shown to correlate with complex cognitive scales such as the Scholastic Achievement Test (SAT) or the American College Test (ACT) [22]. CRT consists of three puzzle-like questions with the intuitive answer being wrong, such as “If it takes 5 machines 5 min to make 5 widgets, how long would it take 100 machines to make 100 widgets?” Participants were answering into a text field. The total score was calculated as the sum of correct answers (therefore, ranging from 0 to 3). The Czech translation of the original test was confirmed by back translation and the assessment of internal reliability has proven good internal consistency (Cronbach alpha = 0.73).

The bullshit receptivity (BSR) was measured by a shortened version of the Bullshit Receptivity Scale [9]. BSR consists of a collection of meaningless computer-generated phrases containing profound words (such as “Wholeness quiets infinite phenomena”) that are rated by the subjects in terms of their profoundness on a 5-point Likert scale (ranging from “not profound” to “very profound”). Given limited space and the arbitrary origin of the items, only the first 5 items of the original scale were selected for the present research, resulting in suboptimal, yet acceptable, internal consistency (Cronbach alpha = 0.63). The total score was calculated as a mean of answers to the specific items. The Czech translation of the original scale was confirmed by back translation.

The digital health literacy (EHEALS) was measured by the eHealth Literacy Scale (eHEALS) [23], a self-reporting questionnaire containing eight claims related to digital health literacy (e.g., “I know how to use the health information I find on the Internet to help me”) to assess on 5-point Likert scales. The total score was calculated as the mean of the individual items. The Czech translation of the original scale was confirmed by back translation, and the assessment of internal reliability revealed excellent internal consistency (Cronbach alpha = 0.92).

### 2.2. Data Collection

Data were collected between 8 and 21 January 2021 (i.e., after the vaccines were introduced in Czechia and before they were made available to the general population) from a convenience sample of students at the universities located in Pilsen, Czechia. Students were delivered a link to an online questionnaire presented via Google Forms by their lecturers. The questionnaire consisted of the above-described scales as part as a longer questionnaire related to their health beliefs. To avoid any possible effects of priming or self-stylization with respect to COVID-related beliefs that might possibly influence responses to the DES and CMQ, the questions regarding COVID-19 were placed at the end of the questionnaire and coronavirus was not mentioned in the introduction of the aims of the research. The study was approved by the Ethical Committee of the University Hospital and Faculty of Medicine in Pilsen (No. 49/2021).

### 2.3. Participants

The sample consisted of 866 university students (mean age 23.58 years; 621 females) of medicine, pedagogy and law. Out of the original 914 responses, 7 participants were excluded as they did not belong to the studied population, and 40 submissions were excluded as duplicates. Considering the small likelihood of participants answering all questions in the same manner, including those requiring a text input, we excluded all duplications. A closer inspection revealed that some participants were answering the survey together in a group and some participants re-submitted the survey with the same answers once informed that the whole year-class was to be exempted from writing an essay based on a specified number of responses.

### 2.4. Statistical Analysis

The scores of individual scales were calculated as the sums of all items for CRT and as means for DES, BSR, CM, eHEALS and (converting the reverse-scored questions) HOAX and CREATED. In CRT, inputs not containing any answer were interpreted as lack of effort rather than lack of ability to solve the puzzle and labeled as missing values rather than incorrect answers. Forced entry multiple linear regression analysis was used to evaluate the effect of the independent variables on conspiracy mentality and belief in specific conspiracy theories, using the function “lm” with its predefined parameters. For testing interactions, moderation analysis was used as described by Wu and Zumbo [24]. To evaluate whether a variable M might be moderating the effect of an independent variable I on a dependent variable D, I and D were centered, scaled and entered into a linear regression model as predictors of M. Then, a second model was created by entering the product of D and I into the first one, and the variance explained in both models was compared using ANOVA. A significant increase of variance explained by adding the product of the two predictors indicated a moderating effect of M. To examine the expected mediating effect of conspiracy mentality and belief in conspiracy theories, mediation analysis was used. After standardizing the data, the respective regression coefficients were calculated, using the “mediate” function, as described by Revelle [25]. Bootstrapping (5000 iterations) was used to estimate confidence intervals. The analysis was conducted in R 3.6.3 [26], using the packages tidyverse [27], psych [28] and QuantPsyc [29].

## 3. Results

### 3.1. Descriptive Statistics

In the sample of 866 university students, 138 (15.94%) considered the probability that COVID-19 is a hoax as being higher than 50%, the mean estimated probability of COVID-19 being a hoax was 16.67%, 149 students (17.21%) considered the probability that COVID-19 was intentionally created by humans to be higher than 50% and the mean estimate was 26.67%. Further descriptive values are depicted in Table 1. As shown in Table 2, all the scales used had at least acceptable reliability, especially taking to consideration the low number of items of some scales. Table 2 also contains a correlation matrix.

### 3.2. Predictors of Conspiracy Beliefs (H1–H4)

Multiple linear regression analysis was used to test whether DES, BSR and CRT predict conspiracy mentality. The results indicated that the three predictors explain 15.52% of the variance (raw value; adjusted = 15.22%; R^2 = 0.16; *F*(3,838) = 51.33, *p* < 0.001). As shown in Table 3, conspiracy mentality was predicted by all three independent variables: DES (β = 0.29, *p* < 0.001), BSR (β = 0.16, *p* < 0.001) and CRT (β = −0.14, *p* < 0.001), supporting the hypothesis 1.

Multiple linear regression analysis was used to test whether DES, eHEALS, BSR and CRT predict beliefs in conspiracy theories about COVID-19, namely, the belief that COVID-19 is a hoax (HOAX) and the belief that COVID-19 is human-made (CREATED). The results summarized in Table 3 indicate that the four predictors explain 7.32% of the variance of HOAX (raw value; adjusted = 6.88%; R^2 = 0.073, *F*(4,837) = 16.52, *p* < 0.001) and 9.62% of the variance of CREATED (raw value; adjusted = 9.19%; R^2 = 0.10; *F*(4,837) = 22.29, *p* < 0.001). HOAX was predicted by DES (β = 0.11, *p* = 0.001), eHEALS (β = −0.12, *p* < 0.001) and CRT (β = −0.18, *p* < 0.001), but not by BSR (*p* = 0.39). CREATED was predicted by all four independent variables: DES (β = 0.12, *p* = 0.001), eHEALS (β = −0.09, *p* = 0.007), BSR (β = 0.09, *p* = 0.005) and CRT (β = −0.22, *p* < 0.001), supporting the hypothesis 2, except for the link between bullshit receptivity and belief that COVID-19 is a hoax.

The results indicate that for individuals, an increase of one standard deviation in dissociative experience is linked with 2.66% higher subjective estimation of probability that coronavirus is a hoax and 2.61% higher subjective probability that coronavirus is human-made. Similarly, an increase of one standard deviation in cognitive reflection is linked with 4.36% lower subjective probability that COVID-19 is a hoax and with 4.79% lower subjective probability that COVID-19 is human-made, and a one standard deviation increase in bullshit receptivity is linked with 1.96% higher subjective probability that coronavirus is human-made. Finally, an increase of one standard deviation in digital health literacy is linked with 2.91% lower subjective probability that COVID-19 is a hoax and 1.96% lower subjective probability that COVID-19 is human-made.

Mediation analysis was used to test whether the effects of personal variables (DES, BSR and CRT) on belief in COVID-related conspiracy theories (HOAX and CREATED) were mediated by conspiracy mentality (CM). The effect of DES on HOAX was fully mediated by CM, with the standardized indirect effect of (0.32) × (0.29) = 0.09 and bootstrapping confirming significance (β = 0.09, CI = [0.07; 0.12]); the unstandardized direct indirect effect was 0.17 (CI = [0.12; 0.23]), while the direct effect did not reach significance (*p* = 0.29). The effect of DES on CREATED was partly mediated by CM, with the standardized indirect effect of (0.32) × (0.41) = 0.13 and bootstrapping confirming significance (β = 0.13, CI = [0.10; 0.17]); the unstandardized direct indirect effect was 0.22 (CI = [0.17; 0.28]), with the direct effect being weak but significant (*B* = 0.05, *p* = 0.048). The effect of BSR on HOAX was not found significant in the previous regression model. The effect of BSR on CREATED was fully mediated by CM, with the standardized indirect effect of (0.20) × (0.41) = 0.08, bootstrapping confirming significance (β = 0.08, CI = [0.05; 0.11]); the unstandardized direct indirect effect was 2.36 (CI = [1.51; 3.25]), while the direct effect did not reach significance (*p* = 0.52). The effect of CRT on HOAX was partly mediated by CM, with the standardized indirect effect of (−0.19) × (0.28) = −0.05, bootstrapping confirming significance (β = −0.05, CI = [−0.08; −0.03]); the unstandardized direct indirect effect was −1.09 (CI = [−1.56; −0.66]) and the direct effect was –3.03 (*p* < 0.001). The effect of CRT on CREATED was partly mediated by CM, with the standardized indirect effect of (0.20) × (0.41) = −0.08, bootstrapping confirming significance (β = −0.08, CI = [−0.05; −0.10]); the unstandardized direct indirect effect was −1.38 (CI = [−1.89; −0.90]) and the direct effect was −3.11 (*p* < 0.001). The results indicate that while most of the effect of dissociation and bullshit receptivity on beliefs in COVID-related conspiracy theories is mediated by conspiracy mentality, cognitive reflection is affecting COVID-related conspiracy beliefs through its link to conspiracy mentality as well as directly. Thus, our data support the hypothesis 3.

Moderation analysis was used to test whether the effect of dissociation experience (DES) on belief in COVID-related conspiracy theories (HOAX and CREATED) may be reduced by digital health literacy (eHEALS). Standardized eHEALS and HOAX or CREATED were entered into the first step of a regression model. The second step also contained the product of eHEALS and HOAX (or CREATED), and both regression models were compared with ANOVA. The result has shown no significant difference in the variance explained (*F*(1,862) = 1.96, *p* = 0.16 for HOAX; *F*(1,862) = 2.11, *p* = 0.14 for CREATED), indicating that the effect of DES on COVID-related conspiracy beliefs is not affected by digital health literacy. Thus, our data do not support the hypothesis 4.

### 3.3. Moderation of the Link between eHEALS and Conspiracy Beliefs (H5–H6)

Moderation analysis was used to test whether the correlation between how respondents answer the eHEALS questionnaire and their belief in COVID-related conspiracy theories (HOAX and CREATED) may be moderated by cognitive reflection (CRT). eHEALS, HOAX and CREATED were scaled and centered and eHEALS and HOAX or CREATED were entered in the first step of the regression analysis. The product of eHEALS and HOAX or CREATED respectively, was then added to the second step of regression analysis. Comparing both models with ANOVA revealed that the product of HOAX and eHEALS increased the variance explained by the model (*F*(1,838) = 5.21, *p* = 0.02), indicating a moderation, while the product of CREATED and eHEALS has not increased the explained variance significantly (*F*(1,838) = 0.68, *p* = 0.41).

To examine the moderating effect of CRT on the link between eHEALS and HOAX, regression models of eHEALS predicting HOAX were calculated separately for participants with above average (i.e., 2 or more correct answers to the 3 CRT questions) and below average (i.e., 1 or less correct answer) CRT score. As depicted in Figure 1, in those achieving above-average CRT score, high eHEALS predicted lower values of HOAX (β = −0.24, *p* < 0.001), in those scoring below average in CRT, HOAX was not linearly dependent on eHEALS (β = −0.05, *p* = 0.36). Given that the average HOAX score was below the middle answer (m = 0.24 on a scale ranging from 0 to 1), a moderating effect of CRT might result from inattention of participants: those achieving lower CRT score might have been answering the questionnaire with less care, being more likely to choose random answers on the HOAX scale, therefore increasing the HOAX scores independently of eHEALS. However, since the average eHEALS is, on the other hand, above the middle answer (m = 0.39 on a scale ranging from 1 to 5), such a group of participants with above-average HOAX scores and below-average eHEALS scores would rather strengthen than disrupt the negative correlation. Additionally, the reliabilities of eHEALS (Cronbach alpha = 0.91) and HOAX (Cronbach alpha = 0.89; including one reverse-scored answer) are almost identical in the group with lower CRT as in the whole sample. Thus, the results support our hypothesis 5 with respect to the belief that COVID is a hoax.

The same procedure was followed to test whether the correlation between how respondents answer the eHEALS questionnaire and their belief in COVID-related conspiracy theories (HOAX and CREATED) may be moderated by their bullshit receptivity (BSR). Comparisons of the respective regression models using ANOVA have not revealed any significant moderation effect neither in case of HOAX (*F*(1,862) = 0.42, *p* = 0.52), nor in case of CREATED (*F*(1,862) < 0.001, *p* = 0.98), therefore not supporting our hypothesis 6.

## 4. Discussion

The presented study shows that high experience with dissociation and low cognitive reflection predict higher belief in conspiracy theories about COVID-19, and that this relationship is mediated by conspiracy mentality. Beliefs in COVID-related conspiracy theories were further related to lower digital health literacy and, to some extent, bullshit receptivity. The effects are, however, weak, explaining only 7–10% of the variance in COVID-related conspiracy beliefs. Additionally, the well-established relationship between digital health literacy measured by eHEALS scale and belief in COVID-related conspiracy theories was moderated by cognitive reflection, supporting warnings that the subjective health literacy scales might be biased when used to measure interindividual differences.

Our data support the hypothesis that people prone to dissociation are also more susceptible to conspiracy theories in general as well as to those related to COVID-19. Our results are consistent with a previous study showing strong correlation between dissociation and conspiratorial beliefs [30] as well as findings that both conspiracy theories [31,32] and dissociation [33] are related to paranormal thinking. The experience with dissociation was a stronger predictor of conspiracy mentality than cognitive measures (cognitive reflection and bullshit receptivity). Its effect might be even larger in the general population than in our student sample, given that the effect of paranormal thinking on belief in conspiracy theories is reduced by education [34]. The same effect might have contributed to our modest effect of dissociation on conspiracy theories about COVID-19.

Digital health literacy was related to lower belief in conspiracy theories, matching some of the previous results [12], while contradicting others [16]. The negative link between digital health literacy measured by the eHEALS scale and belief in COVID-related conspiracy theories only held in persons with high cognitive reflection, but not bullshit receptivity. These results are in line with our expectation that certain groups of people might overestimate their skills when answering self-reporting questionnaires such as eHEALS. Because overestimating one’s own understanding of complex relations is related to belief in conspiracy theories in politics [14], overestimating one’s understanding of health-related information might increase both belief in conspiracy theories about COVID-19 as well as positive bias in self-assessment in the eHEALS questionnaire. This might help to explain the unexpected positive correlation between eHEALS score and conspiracy beliefs [16,17] or the negative link between some aspects of health literacy and vaccination compliance [18]. As a result, the research using subjective measures to estimate digital health literacy (including the presented one) may systematically underestimate the positive impact of digital health literacy on belief in conspiracy theories, willingness to get vaccinated or other outcome measures. The biased measurement of health literacy might also have caused the absence of the expected interaction of the effects of health literacy and dissociation on COVID-related conspiracy beliefs.

Consistent with previous research, conspiracy mentality and COVID-related conspiracy theories were related to low cognitive reflection and high bullshit receptivity, except for the belief that coronavirus is a hoax, which was found to be independent of bullshit receptivity. People with less analytical cognitive style measured by lower cognitive reflection were shown to be more prone to conspiracy theories about COVID-19 in previous research [7,8] and to follow more untrustworthy information sources on Twitter, which may further increase their exposure to conspiracy theories [35]. While bullshit receptivity was repeatedly shown to be related to belief in generic conspiracy theories [9,10,36], it might not be a strong predictor of COVID-related conspiracy beliefs, perhaps due to the intensive public debate that may make individual attitudes to COVID-19 less dependent on the ability to distinguish meaningful from meaningless statements.

The beliefs in COVID-related conspiracy theories depend on the described variables to a lesser extent than the general conspiracy mentality. Altogether, experience with dissociation, cognitive reflection and bullshit receptivity explained 16% of the variance of conspiracy mentality, while together with digital health literacy, these variables only explained 7% of the variance of the belief that COVID-19 is a hoax and 10% of variance of the belief that it is human-made. Considering that conspiracy theories are, besides psychological factors, related to epistemic motives and political factors [37], the lower impact of psychological variables on conspiracy theories related to coronavirus might reflect the politicization of the pandemic and its strong impact on individual lives. Stressful events increase the tendency for motivated reasoning, increasing motivation to adjust beliefs to preferred views at the expense of matching reality [38,39,40]. Thus, the burden of pandemics and related restrictions might have increased susceptibility to conspiracy beliefs in those most affected or those most predisposed to motivated reasoning, irrespective of their general conspiracy mentality.

### 4.1. Limits

Only two conspiracy theories about COVID-19 were used for the analysis, limiting its generalizability to the whole scope of conspiracy beliefs about coronavirus. The timing of the data collection during the pandemic of coronavirus, in the weeks after the vaccination campaign was started, limits generalization of the absolute numbers in time. The immediate effect of the pandemic on individual lives may have strengthened the political and epistemic predictors of the conspiratorial explanations on the expense of the psychological ones, as suggested previously [2]. The sample of university students may have influenced the effects of the scrutinized factors on the beliefs in conspiracy theories. Because education was shown to undermine the reasoning processes behind conspiracy theories [33], it might have reduced the effect of dissociation. The variance of cognitive reflection and digital health literacy may be lower in our sample of university students compared to the general population, which might have reduced the scope of their effect on beliefs in conspiracy theories. At the same time, homogeneity of our sample in terms of education and social status might have accented the differences in the psychological and cognitive predispositions. Geographic generalization may be limited by the specifics of the political and social situation in Czechia. Cross-sectional design of our study has obvious limitations for deducing causality. Finally, our hypothesis concerning possible bias in the eHEALS scale in respondents with low cognitive reflection was only confirmed with respect to one of the two conspiracy theories.

### 4.2. Future Research

The link between conspiracy theories and dissociation experience in healthy students warrants exploration of a possible effect of clinically relevant dissociation on conspiracy beliefs. Considering our finding that the relationship between eHEALS score and belief in conspiracy theories depends on cognitive reflection, it should be further studied whether and to which extent subjective measures of digital health literacy may be affected by lack of cognitive reflection. Research should further address differences in predictors of different conspiracy theories. First, our data revealed a difference in the effect of bullshit receptivity on two conspiracy theories on a similar topic. Similarly, collective control was positively associated with most conspiracy beliefs about COVID-19, but negatively associated with those beliefs that blame the government in Poland [41]. Second, our data revealed that COVID-related conspiracy beliefs depend on psychological predictors less than general conspiracy mentality, perhaps because conspiracy beliefs about COVID-19 are influenced by situational variables (i.e., personal losses, attitudes to the institutions reacting to the crisis) more than conspiracy theories explaining distant events, such as the deaths of JF Kennedy or Princess Diana. Indeed, the effects of politically motivated reasoning on COVID-related conspiracy theories were larger compared to the effects of anxiety, desire for certainty or cognitive reasoning [2], and national narcissism had a large effect on belief in various COVID-related conspiracy theories [42]. Further, variables related to political orientation as well as those related to education and access to information may explain belief in conspiracy theories about coronavirus to the same or higher extent as psychological constructs [43], with institutional trust affecting COVID-related misbeliefs twice more than digital health literacy [12], altogether warranting further research on the difference between psychological and political predictors of different conspiracy theories.

### 4.3. Implications

Promotional campaigns against COVID-related conspiracy theories should be presented in a way that is accessible especially to the individuals vulnerable to conspiracy theories. As their characteristics involve low cognitive reflection and high bullshit receptivity, promotional messages should address intuitive rather than analytical thinking. Findings depending on subjective measures of digital health literacy together with variables that may be correlated with cognitive reflection should be taken with caution.

## 5. Conclusions

Belief in conspiracy theories about coronavirus is an important predictor of nonadherence to the measures to reduce the impact of the COVID-19 pandemic [3,5]. Our data from Czech university students indicate that COVID-related conspiracy theories are linked to high dissociation tendencies, low cognitive reflection, low digital health literacy and, in part, high bullshit receptivity. At the same time, the predictors only explained 7–10% of the variance of COVID-related conspiracy theories and 16% of conspiracy mentality, suggesting that other, perhaps socio-political factors, may be needed to fully understand the origins of conspiracy theories. The effect of self-perceived digital health literacy on one of the conspiracy beliefs was only present in respondents with high cognitive reflection, indicating that self-perceived scales may inflate the level of digital health literacy in subjects with low cognitive reflection.

## Figures and Tables

**Figure 1 ijerph-18-05065-f001:**
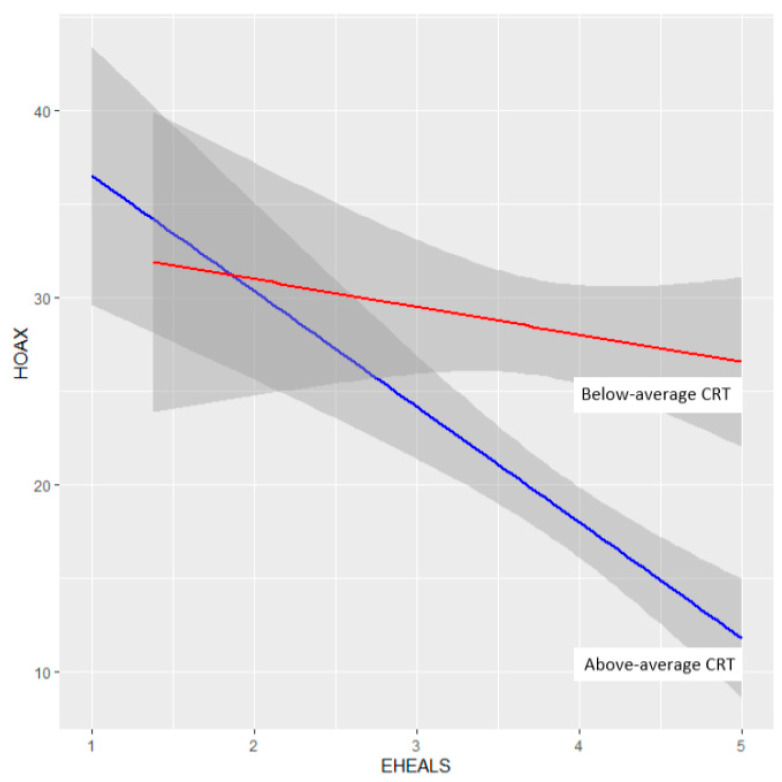
Comparison of correlations between the eHEALS score (eHEALS) and the belief that COVID-19 is a hoax (HOAX) based on score in the cognitive reflection test (CRT). In participants with above-average CRT scores (in blue), the belief that COVID-19 is a hoax (axis Y) decreases with increasing digital health literacy (axis X), while in participants with below-average CRT score (in red), digital health literacy and belief that COVID-19 is hoax are not linearly related.

**Table 1 ijerph-18-05065-t001:** Descriptive statistics.

	n	Min	Max	Mean	Med	Standard Deviation	Standard Error	Skew	Kurtosis
DES	866	0	78.21	17.58	14.11	13.1	0.45	1.24	1.53
CM	866	2	100	56.04	58	20.21	0.69	−0.15	−0.47
HOAX	866	0	100	23.60	16.67	24.22	0.82	0.95	0
CREATED	866	0	100	29.77	26.67	21.76	0.74	0.66	−0.11
eHEALS	866	1	5	3.85	4	0.82	0.03	−0.66	0.07
CRT	842	0	3	1.51	2	1.19	0.04	−0.05	−1.52
BSR	866	1	5	2.53	2.6	0.76	0.03	0.17	−0.26

DES—Dissociation Experience Scale, CM—Conspiration Mentality, HOAX—conspiracy theory that COVID-19 is a hoax, CREATED—conspiracy theory that COVID-19 is human-made, eHEALS—digital health literacy, CRT—cognitive reflection test, BSR—bullshit receptivity scale.

**Table 2 ijerph-18-05065-t002:** Correlation matrix, Cronbach alpha and number of items in scales.

	No. of Items	Cronbach Alpha	DES	CM	HOAX	CREATED	eHEALS	CRT
DES	28	0.93	–					
CM	5	0.82	0.33	–				
HOAX	3	0.88	0.15	0.30	–			
CREATED	3	0.67	0.17	0.42	0.46	–		
eHEALS	8	0.92	−0.08	−0.06	−0.14	−0.11	–	
CRT	3	0.73	−0.16	−0.19	−0.21	−0.25	0.05	–
BSR	5	0.63	0.13	0.21	0.06	0.12	−0.05	−0.05

DES—Dissociation Experience Scale, CM—Conspiration Mentality, HOAX—conspiracy theory that COVID-19 is a hoax, CREATED—conspiracy theory that COVID-19 is human-made, eHEALS—digital health literacy, CRT—cognitive reflection test, BSR—bullshit receptivity scale.

**Table 3 ijerph-18-05065-t003:** Linear regression models of conspiracy mentality (top), the belief that coronavirus is a hoax (middle) and the belief that coronavirus is human-made (bottom).

	*B*	Standard error	Beta	*p*	95% CI
CM ~ DES + BSR + CRT					
(Intercept)	40.65	2.53		<0.001	35.67–45.62
DES	0.44	0.05	0.29	<0.001	0.34–0.54
BSR	4.37	0.86	0.16	<0.001	2.69–6.05
CRT	−2.36	0.55	−0.14	<0.001	−3.44–−1.29
HOAX ~ DES + EHEALS + BSR + CRT					
(Intercept)	36.94	5.00		<0.001	27.14–46.75
DES	0.20	0.06	0.11	0.001	0.08–0.32
eHEALS	−3.64	0.98	−0.12	<0.001	−5.56–−1.72
BSR	0.91	1.06	0.03	0.39	−1.17–2.99
CRT	−3.61	0.68	−0.18	<0.001	−4.94–−2.28
CREATED ~ DES + EHEALS + BSR + CRT					
(Intercept)	34.53	4.48		<0.001	25.73–43.33
DES	0.20	0.06	0.12	<0.001	0.09–0.31
eHEALS	−2.36	0.88	−0.09	<0.01	−4.08–−0.64
BSR	2.64	0.95	0.09	<0.01	0.78–4.51
CRT	−3.97	0.61	−0.22	<0.001	−5.16–−2.78

B, beta—unstandardized and standardized regression coefficient, 95% CI—confidence interval; DES—Dissociation Experience Scale, CM—Conspiration Mentality, HOAX—conspiracy theory that COVID-19 is a hoax, CREATED—conspiracy theory that COVID-19 is human-made, eHEALS—digital health literacy, CRT—cognitive reflection test, BSR—bullshit receptivity scale.

## Data Availability

The data presented in this study are available upon request.

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
