# Peer review of "Dissociation, Cognitive Reflection and Health Literacy Have a Modest Effect on Belief in Conspiracy Theories about COVID-19"

_ijerph, 2021, doi:10.3390/ijerph18105065_

Round 1
Reviewer 1 Report
Thank you for giving me the possibility to review the manuscript "Psychology losing grip of politized conspiracies: Dissociation, cognitive reflection and health literacy have modest effect on belief in conspiracy theories about COVID-19”, submitted to the International Journal of Environmental Research and Public Health.
The manuscript reports a cross-sectional study on the psychological and cognitive characteristics influencing general and Covid-related conspiracy beliefs, in a large sample of Czech university students. The topic is timely and relevant. However, there are a few suggestions I wish to make, before granting the acceptance. I list my suggestions below:
- I feel that the title does not fully represent the content of the manuscript; I think that the Authors could better “sell” their research paper by pointing out their main findings, that is the effect of experience with dissociation and cognitive reflection on beliefs in Covid-related conspiracy beliefs.
- About the Bullshit receptivity scale (BSR): I understand that this is the name of the original scale (Pennycook et al., 2015), but I would prefer to not read those words on a published scientific article; would it be possible to replace it with “Rubbish receptivity scale” or “Non-sense receptivity scale”? those words were used by Pennycook et al. themselves.
- About the eHEALS scale: I agree with the Authors that studying bias in self-report measures is of extreme relevance to put into perspective the results of our researches; however, also “objective” measure of scientific literacy have shown inconsistent results; for instance, results on the relationship between scientific literacy and attitudes towards biotechnology are somewhat contradictory, with many studies pointing to a weak (or no) correlation between knowledge of scientific facts and a positive attitude to science. For instance:
- Allum N, Sturgis P, Tabourazi D, Brunton-Smith I. Science knowledge and attitudes across cultures: A meta-analysis. Public Understanding of Science 2008; 17(1): 35-54, DOI: 10.1177/0963662506070159.
- Bauer MW, Durant J, Evans G. European public perceptions of science. International Journal of Public Opinion Research 1994; 6(2): 163–186, DOI: https://doi.org/10.1093/ijpor/6.2.163.
- Miller JD. Public understanding of, and attitudes toward, scientific research: What we know and what we need to know. Public Understanding of Science 2004; 13: 273-294, DOI: 10.1177/0963662504044908.
- Pivetti, M., Caggiano, A., Cieri, F., Battista, S. D., & Berti, C. (2017). Support for the Forensic DNA Database and Public Safety Concerns: An Exploratory Study. The Open Psychology Journal, 10(1).
My point is that maybe scientific literacy, being subjectively or objectively measured, has been found to be non-related to positive attitude towards scientific issues nor conspiracy beliefs; I personally find a bit devious the idea put forward by the Authors on p. 2 (lines 77-92);
- As for the statistical analysis: on p. 4, line 186 states “vaccination intentions” but I have found no data on vaccination intentions in the manuscript; is this a typo? Also, in the same line, the brackets are empty; is this correct? (I am familiar with SPSS, not with R); line 196: the brackets are empty;
- Table 2 (p. 5) is not commented in the text; also, asterisks indicating significance level are missing in the Table (e.g. .33**); and please, the put significance level under the table, e.g. “** p < .001” following APA guidelines;
- To me, all footnotes should be placed in text;
- About the results on p. 6: I wonder if it is possible to add conspiracy mentality (CM) as independent variable in the regression on the two (HOAX an CREATED) Covid-related conspiracy theories, along with DES, EHEALS, BSR AND CRT; my point is that also conspiracy mentality could affect the belief in Covid-related conspiracy theories; also the Authors could consider using structural equation model to analyse their data and add this analysis to the manuscript;
- About the moderation of the link between EHEALS and HOAX conspiracy beliefs (p. 7-8, lines 292-321): I wonder if an alternative explanation of the moderation would be that participants with lower CRT score would not be able to hypothesize that Covid is a HOAX (for instance the word “mislead” would be difficult to understand for them), while they would be able to say that it is created by humans. In this sense, participants with lower CRT were not responding with less care but they were not able to grasp the meaning of the sentences in the HOAX scale (e.g. functional illiteracy).
Anyway, I think that the paper would benefit if the (grounded) criticism on the EHEALS scale would be placed in the Study limitation section, and not integrated into the manuscript as one of the objective/hypothesis of the paper.
- In the Discussion: line 373-374: would it be “were related with LOW (not high) cognitive reflection and HIGH (not low) bullshit receptivity…”;
Line 383: “on or”; maybe “or” is a typo?7
Line 392: “politization”; would it be “politicization”?
Line 393-395: it is not fully clear to me.
Author Response
Please, see the attachment.

Reviewer 2 Report
The article makes an important contribution to our understanding of some cognitive predictors of conspiracy mentality and beliefs in conspiracy theories.
The theoretical introduction and the methodological section are well presented.
However, I suggest special attention to the presentation of the results, particularly in section 3.2 where the models obtained from linear regression explaining Conspiracy Mentality (CM) and COVID-related conspiracy belief theories (HOAX and CREATED scales) are illustrated. The R-square values presented explain a very modest amount of the variance of the dependent variables, 15.6% for CM and 7.3% and 9.6% for HOAX and CREATED. There is therefore a large amount of variance (84.4%, 90.4% and 92.7% respectively) that needs to be explained by introducing variables that have not been taken into account in the models presented in this paper. I therefore suggest that the authors present these results with some caution, pointing out that the results obtained explain only a very modest amount of the variance of the variables examined and that additional constructs should be considered to explain Conspiracy Mentality (CM) and COVID-related conspiracy belief theories (HOAX and CREATED).
Author Response
Please, see the attachment.
